# A Review of the Gate-All-Around Nanosheet FET Process Opportunities

**Sagarika Mukesh** [†,‡] (ID) **and Jingyun Zhang** [*,‡]

IBM Research Albany, Albany, NY 12203, USA
* Correspondence: zhangji@us.ibm.com
† Current address: 257 Fuller Road, Suite 3100, Albany, NY 12203, USA.
‡ These authors contributed equally to this work.

**Abstract:** In this paper, the innovations in device design of the gate-all-around (GAA) nanosheet FET are reviewed. These innovations span enablement of multiple threshold voltages and bottom dielectric isolation in addition to impact of channel geometry on the overall device performance. Current scaling challenges for GAA nanosheet FETs are reviewed and discussed. Finally, an analysis of future innovations required to continue scaling nanosheet FETs and future technologies is discussed.

**Keywords:** gate-all-around nanosheet FETs; multi-Vt offerings; bottom dielectric isolation; power-performance improvement; transistor scaling; Moore's Law



## 1. Introduction

Gate-all-around (GAA) nanosheet field effect transistors (FETs) are an innovative next-generation transistor device that have been widely adopted by the industry to continue logic scaling beyond 5 nm technology node, and beyond FinFETs [1]. Although gate-all-around transistors have been researched for many years, the first performance benchmarking on scaled pitch of 44/48 nm CPP (contact-poly-pitch) was presented less than five years ago [2–8]. To fully appreciate the advantages provided by stacked nanosheet gate-all-around transistors, it is important to understand some of the challenges faced by the state-of-the-art FinFETs, and, in general, the trends that have motivated industry wide innovations over the years. Historically, device architecture innovations have been driven by short channel effects (SCEs) that come into play while achieving power performance area (PPA) scaling. SCEs occur when the channel length is on the same order of magnitude as the source-drain depletion layers [9]. Over the years, several innovations, such as the stress technology and high-k metal gate, have enabled scaling [10,11]. FinFETs were the first-ever change in architecture in the history of transistor devices to enable scaling by introducing the trigate control, thereby giving the gate-length scaling another few generations of runtime [12,13]. The gate-all-around nanosheet FETs are only the second time in the history of transistor devices, that a completely different architecture is adopted by the industry.

Scaling FinFETs beyond 7 nm node results in exacerbated SCEs, motivating a move from a tri-gate architecture to a gate-all-around architecture [14]. Among the gate-all-around architectures explored by the semiconductor industry, while the nanowires provided best electrostatic control, wider nanosheets are the ones that provide higher "on" current and improved electrostatic control over FinFETs [15,16]. Figure 1 shows a schematic of a FinFET and a GAA nanosheet FET, where the key components of the two technologies are highlighted. The components that are common between the two technologies include the shallow trench isolation, source/drain epitaxies, and the high-k metal gate; whereas the structural differences include a tri-gate for FinFETs vs. gate-all-around for nanosheets. To achieve an advantage in performance, multiple nanosheets must be stacked on top of each other, unlike FinFETs, where one fin constitutes one device. The channel thickness is lithographically defined for FinFETs, which puts a limit on scaling due to patterning

resolution, whereas this channel thickness (also referred to as $T_{Si}$, thickness of silicon) is defined through epitaxially grown layers of Si on top of epitaxially grown layers of low concentration germanium SiGe, providing superior channel uniformity across wafer, and eliminating process complications.

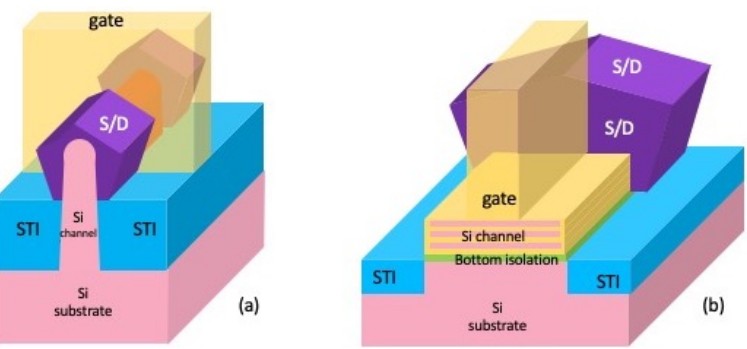

**Figure 1.** This figure shows a FinFET and a GAA nanosheet FET side-by-side. (**a**) A FinFET with shallow trench isolation (STI), source/drain (S/D) epitaxy, and a high-k metal trigate is depicted schematically. (**b**) A GAA nanosheet FET with STI, S/D epitaxy, bottom dielectric isolation (BDI), and high-k all-around metal is pictured. Some features, such as BDI and isolation between gate and S/D, are unique to the GAA nanosheet FETs.

Figure 2 shows a GAA-FET and highlights some of its key features that have been carefully engineered and extensively studied over the last few years. These features include discrete silicon sheets horizontally stacked to form one device, a high-k metal gate filling the space between the silicon channels, bottom dielectric isolation from bulk substrate, lithographically defined sheet width, process controlled gate-length, and inner spacer formation for gate to source-drain isolation. Some aspects of these GAA nanosheet FETs, such as inducing strain to increase hole mobility, have been a hot topic to improve overall device performance, but will not be covered in this paper [17–19]. Other aspects, such as multiple threshold voltage (Multi-$V_T$) options for high power and low power devices, impact of channel geometry on device performance, and integration and impact of full dielectric isolation, are reviewed in this paper [20–24].

The structure of the remaining paper is as follows: Section 2 highlights the key integration modules and shows a high-level process flow; Section 3 covers bottom dielectric isolation—its need, integration, and impact on device performance; Section 4 explores the impact of channel geometry on device performance, especially the impact of channel geometry on the hole mobility; Section 5 discusses different integration approaches for enabling multiple threshold voltages (multi-$V_T$) in GAA nanosheet FETs; Section 6 briefly discusses innovations in interconnects and power delivery networks that are needed to extract the value from scaled nanosheet architecture; and, finally, Section 7 discusses the direction of the transistor industry beyond GAA nanosheet FETs.

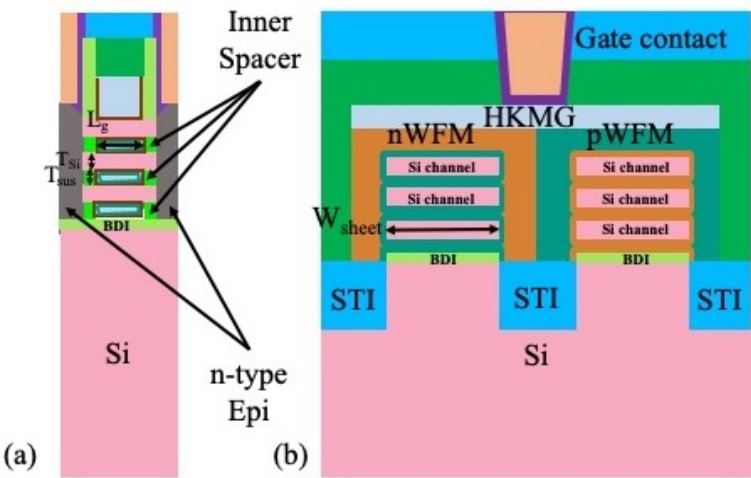

**Figure 2.** This figure shows a schematic for a gate-all-around nanosheet FET, with its key features highlighted. (**a**) shows a cut across the source-drain region where the key features highlighted are the bottom dielectric isolation (BDI), the thickness of the silicon channels ($T_{Si}$), the distance between the silicon channels ($T_{sus}$), and the gate length ($L_g$). The inner spacer, and the n-type epi are also highlighted here. (**b**) shows a cut across the gate region where the key features highlighted are shallow trench isolation (STI), n-type work function metal (WFM), p-type WFM, the high-k metal gate (HKMG), and the sheet width ($W_{sheet}$).

## 2. Integration of Gate-All-Around Nanosheet FETs

The integration of GAA nanosheet FETs involves several novel steps requiring a series of innovations to enable this technology. The key integration modules are listed below: [21]:

1. Stacked nanosheet formation: a stack of SiGe and Si are epitaxially grown on the Si substrate; the thickness of each layer can be controlled with high precision.
2. Fin reveal and STI: the devices are lithographically defined and shallow trench isolation is performed to isolate neighboring devices.
3. Dummy gate formation: a poly silicon dummy gate is formed to enable downstream processing.
4. Inner Spacer and Junction formation: n-type or p-type source/drain epitaxial layers are selectively formed on either sides of the exposed nanosheet ends [25].
5. Replacement metal gate formation:
   - Dummy gate pull: the dummy gate is etched out to reveal a cavity, at the bottom of which nanosheets are located,
   - Sacrificial SiGe channel release: the SiGe channels in between the nanosheets are etched out to enable filling up with high-k metal gate,
   - High-k Metal Gate (HKMG) formation: an interfacial oxide, a high-k dielectric layer, and the n-type or p-type work functions are selectively deposited.

## 3. Full Bottom Dielectric Isolation

In this section, we highlight the comparison between a full bottom dielectric isolation (BDI) and punch through stopper (PTS) scheme as examined by [24]. To introduce the problem, we first introduce the "fat-fin" effect that is unique to GAA nanosheets, where process non-idealities can result in a structure that causes an increased capacitance in the bulk region below the nanosheets, as shown in Figure 3. Although this structure is unique to GAA nanosheets, the effect, also known as sub-fin leakage, exists for FinFETs and is dealt with using the punch through stopper scheme. So, a comparison between this established PTS scheme and novel BDI scheme is performed on the basis of off-state leakage current, short channel effects, and effective capacitance ($C_{eff}$); it is shown that BDI could potentially provide improved $C_{eff}$ and power-performance co-optimization.

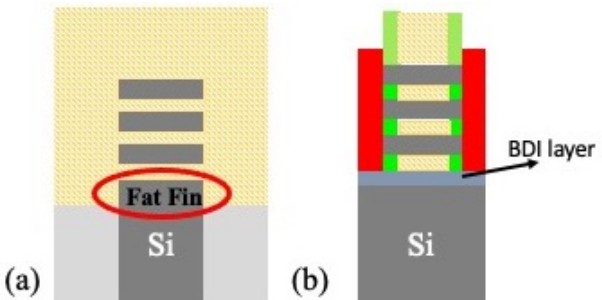

**Figure 3.** (**a**) A figure depicting a cross-fin cut showing high-k metal gate extension beyond the bottom sheet due to poor process control. As the metal depth below the bottom device increases, the performance penalty due to increased $C_{effective}$ also increases. (**b**) A figure showing improved process control due to full bottom dielectric isolation (FBDI) in the source/drain region.

### 3.1. Integration

Integrating a full bottom dielectric isolation entails adding a high-concentration SiGe layer at the bottom of the Si, SiGe nanosheet stack. Adding this layer, and later selectively etching it, requires lowering of Ge concentration in the SiGe layers used for the nanosheet stack. This introduces lower selectivity between Si and SiGe, resulting in Si loss during SiGe channel removal—requiring careful consideration of the stack thicknesses to ensure the $T_{Si}$ is not too thin at the end of the entire process flow. We can see the BDI sitting under the S/D region in Figure 3b.

### 3.2. Experiments

Two splits of the PTS scheme of varying doping concentration are studied along with a full BDI scheme at $V_{ds}$ = 0.7 V for $L_{metal}$ of 12 nm in a 44 CPP device, where their short channel characteristics and power vs. performance is analysed.

### 3.3. Results and Discussion

As seen in Figure 4, full bottom dielectric isolation reduces the off-state leakage current and the DIBL, thereby improving performance and decreasing power. An 18% decrease in power is observed with a 4% improvement in performance between split with and without BDI. The devices with BDI perform better, and they also show better immunity to process variations with respect to sub-channel leakage control. So, full bottom dielectric isolation may be considered as a critical element for enabling a well-performing GAA nanosheet FET.

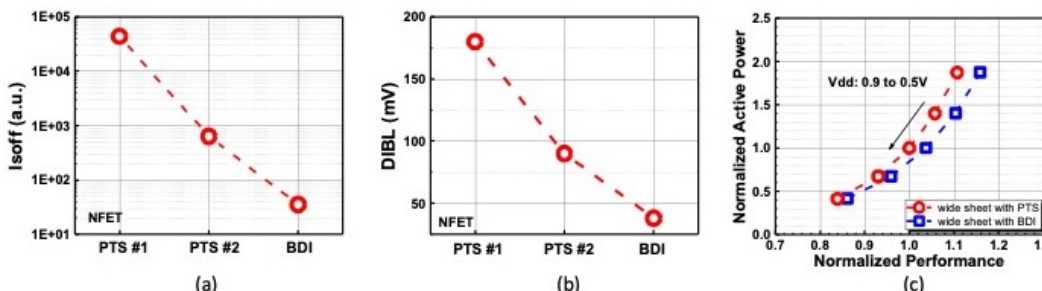

**Figure 4.** This figure captures key performance metrics for GAA FETs using PTS scheme and full BDI. (**a**) $I_{soff}$ extracted from $L_g$ =12 nm devices on PTS and BDI splits. (**b**) DIBL extracted from $L_g$ =12 nm devices on BDI and PTS splits. (**c**) Power vs. performance correlation chart of wide sheet devices for both with and without BDI layer [24].

## 4. Channel Geometry Impact

In this section, the mobility of electrons and holes as a function of channel geometry are studied and 'narrow sheet effect' on carrier transport is observed [23]. $T_{Si}$ is one of the knobs that can enable future $L_g$ scaling needs by improving electrostatic control. Moreover,

the quantization effects for $T_{Si} < 5$ nm becomes severe in SOI and FinFETs, so, it is important to study the same for GAA FETs.

### 4.1. Experiments

Since the mobility of holes ($\mu_h$) is lower for the <100> plane, this plane will dominate hole transport characteristics for pFETs in GAA nanosheet FETs. To study the impact of <100> plane on hole transport, nanosheet devices are fabricated on <100> substrate with <110> transport direction. Figure 5 shows a TEM from the experiments performed, a channel length of 100 nm was chosen for this study. To study the impact of $T_{Si}$ on hole mobility, silicon sheets of different thicknesses were epitaxially grown, and the $T_{Si}$ was measured using TEMs.

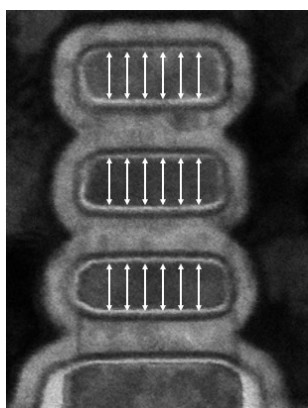

**Figure 5.** A TEM cross-section of GAA nanosheet FETs. The $T_{Si}$ is uniform in thickness along the $W_{sheet}$ direction [23].

### 4.2. Results and Discussion

As seen in Figure 6a, the degradation of $\mu_h$ is attributed to increased phonon scattering with thinner $T_{Si}$. At high fields, as in the case of $N_{inv}$ at $10^{13}$, the mobility is dominated by surface roughness, whereas the peak mobility is primarily impacted by phonon scattering. So, the impact of mobility degradation is more profound for the peak mobility case. However, this degradation of mobility is offset by sheet width $W_{sheet}$ as seen in Figure 6b, which is primarily influenced by the contribution of <100> vs. <110> planes. Wider sheets have more contribution from the <110> plane, resulting in improved mobility, suggesting that both phonon scattering and sheet geometry effects impact hole mobility. Moreover, this dependence on $W_{sheet}$ provides an additional knob for power and performance co-optimization in GAA nanosheet FETs.

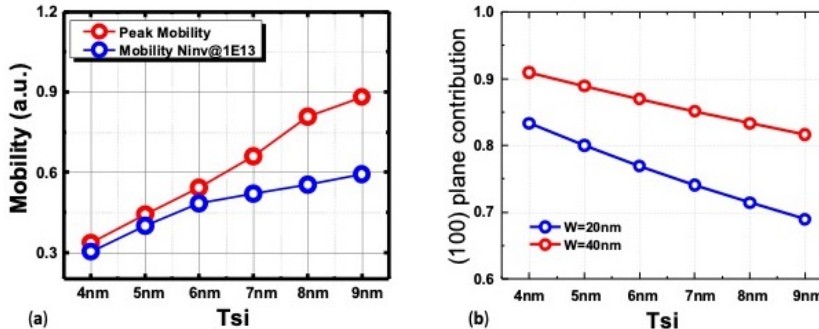

**Figure 6.** (**a**) This plot shows the extracted peak hole mobility and hole mobility for $N_{inv}@10^{13}/\text{cm}^2$ as a function of silicon channel thickness. Degradation in hole mobility is evident for thin sheet values; (**b**) calculated <100> plane contribution to total $W_{eff}$ as a function of $T_{Si}$ (a pure geometrical percentage of the whole nanosheet perimeter) [23].

## 5. Enabling Multiple Threshold Voltages

The ability to co-integrate multiple threshold voltages ($V_T$) is a key requirement for a technology to become an industry standard. Given the unique architecture of GAA FETs, the space for depositing the work function metals is limited as depicted in Figure 7. The replacement metal gate process only leaves the space between the Si channels and inner spacers open—to be filled with the work function metals as per technology requirements. This space, also known as $T_{sus}$ (refer Figure 2), can be controlled by controlling the thickness of SiGe layer grown during the nanosheet stack development module, but is nevertheless highly constrained, and must be engineered carefully to meet the industry standards for device offerings.

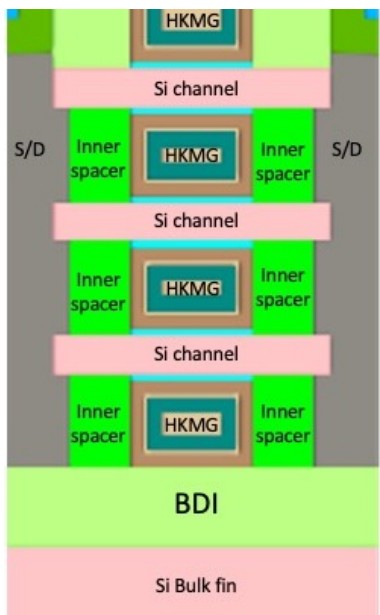

**Figure 7.** This figure shows a close up view of the S/D cross-section. The width of the high-k metal gate here is the gate length $L_g$, and the vertical thickness of this metal gate is determined by $T_{sus}$. Additionally, the inner spacers and bottom dielectric isolation are highlighted.

### 5.1. Integration

Two different approaches are proposed to accommodate multi-$V_T$ offerings in GAA FETs—(1) WFM modification and (2) $T_{sus}$ modification [20]. A process flow overview for WFM modification is presented in Figure 8. One of the challenges highlighted by the integration sequence for $V_T$ modulation is that large $W_{sheet}$ adds process challenges for WFM etch back when WFM is pinched-off between Si channels. To overcome this, ref. [21] proposed filling the space between sheets with a sacrificial material that is easy to etch, selectively opening one of the FETs, and etching away the already deposited work-function metal. This scheme is agnostic of p-type or n-type WFM, and enables both PG (p-FET first) and MY (n-FET first) schemes. This same process can be repeated to achieve different sets of work function metals or to achieve a different stack with more than two WFMs.

The second approach requires changing $T_{sus}$ by changing the channel stack epitaxy thickness during nanosheet formation. A larger space between sheets allows the deposition of a larger volume of work function metal in this space, thereby modulating $V_T$. This design knob is unique to GAA nanosheet FETs when compared with FinFETs, thereby, allowing more design space for Multi-$V_T$ options in these nanosheet FETs.

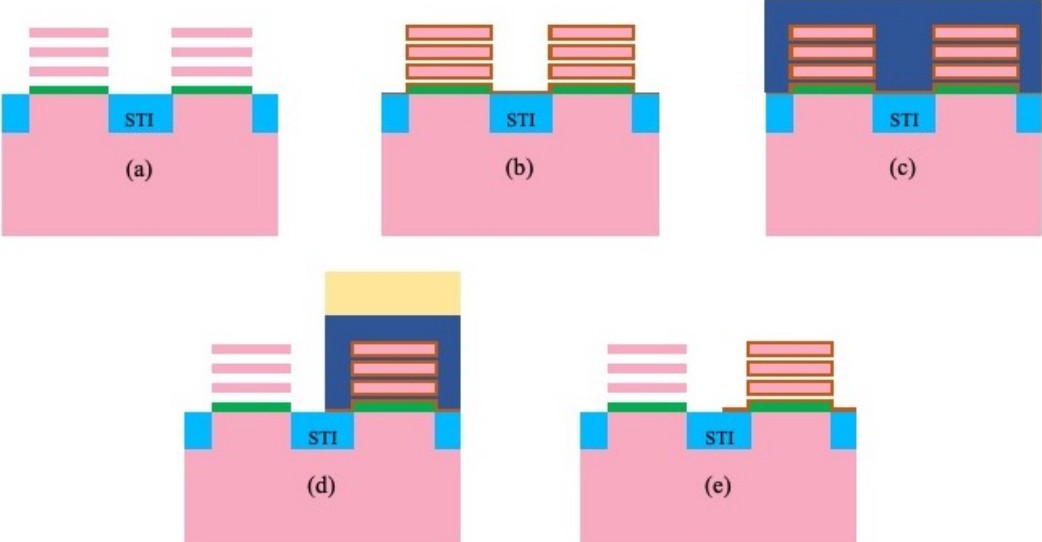

**Figure 8.** An example of $V_T$ modulation as presented in [21] is shown here. (**a**) The gate region post SiGe channel release; (**b**) WFM1 deposition; (**c**) sacrificial material deposition; (**d**) selective patterning and etch of the deposited WFM; and (**e**) removal of patterning stack resulting in a structure with WFM1 along one set of sheets.

Volumeless Multiple Threshold Voltages

A volumeless multi-$V_T$ is a term defined to represent a dipole-based $V_T$ option where a dipole of thickness less than 5 Åis formed, followed by the base work function metals [26,27]. This innovative scheme provides space and gate resistance benefits as shown in the cited references. However, this approach does not directly translate from FinFETs to GAA nanosheet FETs, so dedicated integration of volumeless $V_T$ is proposed in [22]. Moreover, a volumeless $V_T$ also helps with $V_T$ uniformity, which is important for uniform switching of transistors.

### 5.2. Results and Discussions

Several different flavours of $V_T$ are created using novel integration sequence and using the unique design knobs for GAA nanosheet FETs -(a) $T_{sus}$ design; and (b) WFM pinch-off. A dipole-based $V_T$ scheme for nanosheet FETs is also proposed. In addition to these knobs, the $T_{Si}$ design as discussed in Section 4, can be modulated to provide a trade-off between mobility and short-channel effects. So, overall, the GAA nanosheet FETs provide several opportunities for application-based optimization, hence they are suitable for high-power and low-power applications alike.

## 6. Current Challenges

This paper discusses some of the cutting-edge advances in the gate-all-around nanosheet transistor technology over the last five years, and consolidates some of pioneering work in the field. In this section, some of the processing challenges of this technology are covered as reported in the literature. These processing challenges may be broadly categorized into four areas: self-heating, mechanical stability during fabrication, device variability, and Si–SiGe intermixing.

Self-heating effects (SHE) in nanoscale devices result in significant thermal cross talk resulting in device performance degradation [28,29]. Studies have explored novel substrates, such as diamond on silicon to provide improved SHE, but such a scheme is less likely to be adopted in high-volume manufacturing. As such, this problem is open to exploration and solution [30].

An aspect of nanosheet fabrication to carefully consider is the mechanical stability of these sheets during the channel release process. Although nanosheets do allow design

flexibility, aspect ratio of the sheets, and mechanical integrity of the inner spacer play an important role in overall stability of these sheets [31]. Another aspect to optimize is the device variability, which can result from several sources including, but not limited to, line-edge roughness, gate-edge roughness, non-uniform work function metal deposition, and random dopant fluctuations. A recent study analyzes these variability and proposes solution for a complementary GAA nanosheet FET structure [32].

Finally, the initial Si-SiGe stack for nanosheets itself is susceptible to thermal intermixing when going through numerous thermal cycles before the channel release step. There have been several studies examining the extent of this intermixing and the mechanism of such diffusion [33–35]. As long as the SiGe channels can etch selective to Si channel sheets, and the Si sheets are not over-etched due to Si–SiGe intermixing, this effect is tolerable.

## 7. Future Outlook

Although the transistor-level innovation is sufficient to drive the industry forward to the next technology nodes, this section briefly touches upon some innovations in the fields of interconnects and power-delivery for completeness.

An interesting proposal in the field of power delivery is the buried power rail (BPR), which proposes moving the power rails to be located below the transistor devices, thereby, providing area on the front-side for routing flexibility, and to reduce conductor crowding [36,37]. However, such a scheme has a short run-path as the requirement of patterning between the devices will limit contact poly pitch (CPP) scaling. To overcome this limitation, the concept of backside power delivery network (BSPDN) has been proposed, with a recent hardware demonstration of its feasibility [38]. However, this new paradigm brings several technical challenges with it, such as back-side patterning, alignment between the structures on the front-side to those on the back-side, and wafer thinning on the back-side of the wafer. If the industry as a whole decides this is the correct direction, there are tremendous opportunities of innovation for tool vendors and equipment manufacturers to enable this technology at a large scale.

## 8. On the Horizon

Although the industry navigates current challenges to bring the GAA nanosheet FETs to market, researchers are already thinking about what lies beyond nanosheet FETs. The top contenders to continue Moore's law scaling are the Vertical Transport FETs (VTFETs) [39] and stacked transistors [40]. VTFETs change the carrier transport direction from the traditional horizontal direction to vertical direction, thereby relaxing the contraints on scaling barriers, such as gate-length ($L_g$), spacer thickness, and contact size; all of which can be optimized for power or performance, based on the application. Stacked transistors offer a more conventional scaling path by stacking the nFET and pFET transistor over each other, thereby providing area benefit. However, both these technologies present several novel integration and manufacturing challenges, which may be subject of a later review.

Looking beyond the immediate future, there is a large body work on novel materials to enable 2-D transistors [41]. Molybdenum disulfide ($MoS_2$) is one of the top contenders for such technologies with ever improving performance based on mobility, contact resistance, and doping [42]. Graphene is another strong contender for a long time, and literature has been reporting ever improved performance for such transistors over the last decade [43]. Indium oxide is another contender for a wide-gap semiconductor material [44]. Although these technologies are promising, there is an inherent barrier to entry for them due to the large overhead cost of novel equipment for the foundries to enable large scale manufacturing of such transistors. So, silicon based transistors will continue scaling for the coming decades with the growing needs for transistors in existing and new industries.

**Author Contributions:** S.M. prepared the manuscript, J.Z. plotted the results, both authors reviewed the paper. All authors have read and agreed to the published version of the manuscript.

**Funding:** This research received no external funding.



**Data Availability Statement:** Not applicable.

**Acknowledgments:** The authors want to thank their colleagues at IBM Research, Albany for thoughtful discussions.

**Conflicts of Interest:** The authors declare no conflicts of interest.

## Abbreviations

The following abbreviations are used in this manuscript:

| | |
|---|---|
| GAA FETs | Gate-All-Around Field Effect Transistors |
| BDI | Bottom Dielectric Isolation |
| STI | Shallow Trench Isolation |
| WFM | Work Function Metal |
| HKMG | High-k Metal Gate |
| SCE | Short Channel Effects |
| RMG | Replacement Metal Gate |
| PTS | Punch Through Stopper |
| MOL | Middle of Line |
| BEOL | Back End of Line |
| S/D | Source/Drain |
| DIBL | Drain Induced Barrier Lowering |
| TEM | Transmission Electron Microscopy |
| VTFET | Vertical Transport Field Effect Transistors |
| PPA | Power, Performance, and Area |
| BPR | Buried Power Rail |
| BSPDN | Back-Side Power Delivery Network |
| CPP | Contact Poly Pitch |

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
