# Peer review of "A Review of the Gate-All-Around Nanosheet FET Process Opportunities"

_electronics, doi:10.3390/electronics11213589_

Round 1

Reviewer 1 Report

This review paper well provides summary of development of nanosheet FETs. However, considering this paper is a review article, the number of cited references are not sufficient. For example, even thought the nanosheet FET was introduced for the first time by the paper "Three-Dimensional MBCFET as an Ultimate Transistor", IEEE EDL, in 2004, it was not included. 

Moreover, other important concerns with respect to device fabrication or reliability should be discussed on this paper at least brief explanations with related references . For example, 

- Self-heating (Design Optimization of Three-Stacked Nanosheet FET from Self-Heating Effects Perspective, IEEE TED 2022)

- Mechanical stability during fabrication (Inner Spacer Engineering to Improve Mechanical Stability in Channel-Release Process of Nanosheet FETs, Electronics, 2021) 

- Device variability (Impact of Process Variation on Nanosheet Gate-All-Around Complementary FET CFET, IEEE TED 2022)

-Ge inter diffusion on Si/SiGe stacks 

Author Response

This review paper well provides summary of development of nanosheet FETs. However, considering this paper is a review article, the number of cited references are not sufficient. For example, even thought the nanosheet FET was introduced for the first time by the paper "Three-Dimensional MBCFET as an Ultimate Transistor", IEEE EDL, in 2004, it was not included. 

The authors thank the reviewer for their comment. The suggested reference has been added as part of introduction background reference.

[2] Sung-Young Lee et al., "Three-dimensional MBCFET as an ultimate transistor," in IEEE Electron Device Letters, vol. 25, no. 4, pp. 217-219, April 2004, doi: 10.1109/LED.2004.825199.

Moreover, other important concerns with respect to device fabrication or reliability should be discussed on this paper at least brief explanations with related references . For example, 

- Self-heating (Design Optimization of Three-Stacked Nanosheet FET from Self-Heating Effects Perspective, IEEE TED 2022)

- Mechanical stability during fabrication (Inner Spacer Engineering to Improve Mechanical Stability in Channel-Release Process of Nanosheet FETs, Electronics, 2021) 

- Device variability (Impact of Process Variation on Nanosheet Gate-All-Around Complementary FET CFET, IEEE TED 2022)

-Ge inter diffusion on Si/SiGe stacks 

The authors thank the reviewer for their comment. The suggested reference has been added and the manuscript has been revised as follows:

6. Current Challenges

This paper discusses some of the cutting-edge advances in the gate-all-around nanosheet transistor technology over the last five years, and consolidates some of pioneering work in the field. In this section, some of the processing challenges of this technology are covered as reported in the literature. These processing challenges may be broadly categorized into four areas: self-heating, mechanical stability during fabrication, device variability, and Si-SiGe intermixing. Self heating effects (SHE) in nanoscale devices results in significant thermal cross talk resulting in device performance degradation [28,29]. Studies have explored novel substrates such as diamond on silicon to provide improved SHE, but such a scheme is less likely to be adopted in high-volume manufacturing. As such, this problem is open to exploration and solution[30]. An aspect of nanosheet fabrication to carefully consider is the mechanical stability of these sheets during the channel release process. While nanosheets do allow design flexibility, aspect ratio of the sheets, and mechanical integrity of the inner spacer play an important role in overall stability of these sheets [31]. Another aspect to optimize is the device variability, which can result from several sources including but not limited to line-edge roughness, gate-edge roughness, non-uniform work function metal deposition, and random dopant fluctuations. A recent study analyzes these variability and proposes solution for a complementary GAA nanosheet FET structure [32]. Finally, the initial Si-SiGe stack for nanosheets itself is susceptible to thermal intermixing when going through numerous thermal cycles before the channel release step. There have been several studies examining the extent of this intermixing and the mechanism of such diffusion [33–35]. As long as the SiGe channels can etch selective to Si channel sheets, and the Si sheets are not over-etched due to Si- SiGe intermixing, this effect is tolerable.

[28] L. Cai, W. Chen, G. Du, J. Kang, X. Zhang and X. Liu, "Investigation of self-heating effect on stacked nanosheet GAA transistors," 2018 International Symposium on VLSI Technology, Systems and Application (VLSI-TSA), 2018, pp. 1-2, doi: 10.1109/VLSI-TSA.2018.8403821.

[29] Bury, Erik et al. “Self-heating in FinFET and GAA-NW using Si, Ge and III/V channels.” 2016 IEEE International Electron Devices Meeting (IEDM) (2016): 15.6.1-15.6.4.

[30] S. Rathore, R. K. Jaisawal, P. N. Kondekar and N. Bagga, "Design Optimization of Three-Stacked Nanosheet FET From Self-Heating Effects Perspective," in IEEE Transactions on Device and Materials Reliability, vol. 22, no. 3, pp. 396-402, Sept. 2022, doi: 10.1109/TDMR.2022.3181672.

[31] Lee, K.-S.; Park, J.-Y. Inner Spacer Engineering to Improve Mechanical Stability in Channel-Release Process of Nanosheet FETs. Electronics 2021, 10, 1395. https://doi.org/10.3390/electronics10121395

[32] X. Yang et al., "Impact of Process Variation on Nanosheet Gate-All-Around Complementary FET (CFET)," in IEEE Transactions on Electron Devices, vol. 69, no. 7, pp. 4029-4036, July 2022, doi: 10.1109/TED.2022.3176835.

[33] Xia, G. (Maggie); Hoyt, J. L.; Canonico, M. Si–Ge Interdiffusion in Strained Si/Strained SiGe Heterostructures and Implications for Enhanced Mobility Metal-Oxide-Semiconductor Field-Effect Transistors. J. Appl. Phys.

2007, 101 (4), 044901. https://doi.org/10.1063/1.2430904.

[34] Dong, Y. (2014). A systematic study of silicon germanium interdiffusion for next generation semiconductor devices (T). University of British Columbia. Retrieved from https://open.library.ubc.ca/collections/ubctheses/24/items/1.0167516

[35] Thornton, Chappel & Tuttle, Blair & Turner, Emily & Law, Mark & Pantelides, Sokrates & Wang, George & Jones, Kevin. (2022). The Diffusion Mechanism of Ge During Oxidation of Si/SiGe Nanofins. ACS applied materials & interfaces. 14. 29422-29430. 10.1021/acsami.2c05470.

Reviewer 2 Report

See enclosed file.

Author Response

Papers from Industry often report plots with normalized quantities. It is conceivable that some information may be considered confidential but, in some cases, the “secret” has no real value. The hole mobility plot vs. silicon thickness is an example of this kind. A physical property should not be considered confidential information in my view, even if measured in proprietary devices.

The authors thank the reviewer for their insightful comment. Since this is a review paper, no new information can be presented in this article. Moreover, the intent of this plot is to study the sensitivity of mobility vs channel thickness which is critical for device design. The design point will vary depending on the device application.

The manuscript reports that the FET threshold voltage depends upon the distance between stacked nanosheets and suggests a modified distance as a possible approach to multiple threshold voltages. No physical justification for this effect is provided. However, the gate metal surrounding every nanosheet should be able to shield them from mutual interference effects.

The authors thank the reviewer for their insightful comment. The change in distance between the sheets changes the volume of metal in the region --which in turn modulates the threshold voltage. Following text has been modified in the manuscript to convey this.

A larger space between sheets allows the deposition of a larger volume of work function metal in this space, thereby modulating $V_T$.

Fig. 1(a), showing a schematic FinFET, indicates the gate region as S/D rather than G.

The authors thank the reviewer for their comment. The figure has been updated.

The caption of Fig. 6(b) “calculated <100> plane contribution to Weff” is unclear. Is it a pure geometrical percentage of the whole nanosheet perimeter, or is it a simulated contribution of the drain current fraction flowing within the flat region of the nanosheet, with exclusion of the rounded edges? In the former case, the assumption of a current flowing at the periphery of the nanosheet does not consider the QM carrier-confinement effect.

The authors thank the reviewer for their insightful comment. It is a pure geometrical percentage of the whole nanosheet perimeter. Modelling data in reference [24] shows the simulations taking the quantum effects into account ---this study is not highlighted in the review to avoid redundancy.

The reference to Fig. 1 for “Tsus (refer Fig. 1)” should be changed to Fig. 2.

The authors thank the reviewer for their comment, this reference has been fixed.

Figure 8(b), showing an intermediate step of the fabrication process for VT modulation, does not show a uniform WFM1 deposition within the empty spaces between silicon nanosheets. Two of them are in fact colored differently from the remaining four.

The authors thank the reviewer for their comment, this figure has been fixed.

Minor points: (i) “dieelctric” and “usinque” within the caption of Fig. 1 are typos to be corrected; (ii) “enales” should read “enables”; (iii) “hence are suitable” requires a subject: maybe “hence they are suitable”; (iv) “Grpahene” should read “Graphene”.

The authors thank the reviewer for their thorough comment, these errors have been fixed.

Round 2

Reviewer 1 Report

The authors reflected all of the comments by the reviewers in their revised manuscript. I recommend this paper to published.